# Negotiating acceptable termination of pregnancy for non-lethal fetal anomaly: a qualitative study of professional perspectives

Lisa Crowe,[1] Ruth H Graham,[2] Stephen C Robson,[3] Judith Rankin[1]

[1]Institute of Health & Society, Newcastle University, Newcastle upon Tyne, UK
[2]Sociology and Politics, School of Geography, Newcastle University, Newcastle upon Tyne, UK
[3]Institute of Cellular Medicine, Newcastle University, Newcastle upon Tyne, UK

**Correspondence to**
Dr Lisa Crowe;
Lisa.Crowe@ncl.ac.uk

## ABSTRACT

**Objective** This study aims to explore the perspectives of professionals around the issue of termination of pregnancy for non-lethal fetal anomaly (TOPFA).

**Methods** Semi-structured interviews were undertaken with medical professionals (14 consultants in fetal medicine, obstetrics, neonatology and paediatrics) and social care professionals (nine individuals with roles supporting people living with impairment) from the Northeast of England. Analysis adopted an inductive thematic approach facilitated by NVivo.

**Results** The overarching theme to emerge from the interview data was of professionals, medical and social care, wanting to present an acceptable self-image of their views on TOPFA. Professionals' values on 'fixing', pain and 'normality' influenced what aspects of moral acceptability they gave priority to in terms of their standpoint and, in turn, their conceptualisations of acceptable TOPFA. Thus, if a termination could be defended morally, including negotiation of several key issues (including 'fixing', perceptions of pain and normality), then participants conceptualised TOPFA as an acceptable pregnancy outcome.

**Conclusion** Despite different professional experiences, these professional groups were able to negotiate their way through difficult terrain to conceptualise TOPFA as a morally acceptable principle. While professionals have different moral thresholds, no one argued for a restriction of the current legislation. The data suggest that social care professionals also look at the wider social context of a person with an impairment when discussing their views regarding TOPFA. Medical professionals focus more on the individual impairment when discussing their views on TOPFA.

## INTRODUCTION

Termination of pregnancy for fetal anomaly (TOPFA) is legal under the Abortion Act 1967, amended by the Human Fertilisation and Embryology Act (HFEA)[1] with no upper gestational limit if there 'is a substantial risk that if the child was born, it would suffer from such physical or mental abnormalities as to be seriously handicapped' (Clause E). Difficulties have been noted in defining terms

### Strengths and limitations of this study

► This qualitative study provides in-depth data on views on termination of pregnancy for fetal anomaly (TOPFA) from a previously unexplored professional group, social care professionals.
► The qualitative nature of this study allowed for the exploration of a sensitive research topic.
► The use of case studies provided tangible examples with which to explore key issues in the process of negotiating moral acceptability of TOPFA.
► This study was conducted in the Northeast of England, so generalisation cannot be assumed but our conclusions are relevant and applicable in other contexts.

such as 'substantial' and 'serious'.[2–4] In 2016, 3208 terminations were reportedly carried out under Clause E, 2% of the total number.[5] The detection of fetal anomalies is likely to continue to increase due to improvements in fetal imaging and increasing risk factors for fetal anomalies (eg, obesity).[6–9] Fetal anomaly screening is offered to all pregnant women through the National Health Service (NHS) Fetal Anomaly Screening Programme (FASP). The FASP defines a fetal anomaly as an abnormality which 'may indicate the baby might die shortly after birth, conditions that may benefit from treatment before birth, to plan delivery in an appropriate hospital/centre and/or to optimise treatment after the baby is born'.[10] Complex issues emerge as medical professionals juggle multiple moral implications, judge the anomaly in question, offering choice, protecting themselves from prosecution while also providing care to parents.[11] These issues are compounded by the limited in utero treatments available, thus reducing parental options to either TOPFA or continue with the affected pregnancy.[2] Thus, medical professionals working in prenatal diagnosis have the potential to influence

decision-making processes being made about an affected pregnancy; understanding professionals' views of TOPFA is, therefore, of crucial importance. This is especially significant given recent research that has found that care received by parents undergoing TOPFA was felt to not adequately meet their needs; for example, being caught between antenatal and postnatal care settings, yet belonging to neither.[12]

Personal views and experiences affect professional behaviour and the views of professionals impact on the pregnant woman. For example, research suggests that some women have felt they were counselled 'towards' TOPFA,[13 14] and other research has identified variations in counselling techniques.[15 16] Religious affiliation has also been found to impact counselling practices.[17] If personal views impact on counselling practices, this may, in turn, influence decisions about TOPFA. Medical professionals providing TOPFA deal with complex information when deciding whether to offer TOPFA or not.[18 19] However, their knowledge and experience of living with disability and impairment tends to be more limited. This is argued to be of concern if assumptions about experience of impairment lead to disability being automatically equated to 'unhealthy'.[20] Similarly, those in the social care sector, involved in the support and care of those living with impairment, have more knowledge about experiences of impairment, but less insight into the decision-making process that leads to TOPFA. Understanding the views of both medical and social care professionals about TOPFA is important to facilitate the provision of appropriate care and to provide support to those making reproductive decisions, but also to gain enhanced insight into how life with impairment is conceptualised from varied perspectives. Social care professionals have a limited voice in debates on TOPFA. Their views offer a different professional insight into what living with an impairment is like. To our knowledge, there is no research on the opinions of social care professionals on TOPFA.

This study aims to explore the perspectives of professionals around the issue of TOPFA.

## METHODS

This paper reports data collected as part of a larger study examining professionals' views on TOPFA.[21] This paper focuses on the qualitative data. Semi-structured interviews were used to collate data exploring professionals' perceptions of the complex issues surrounding TOPFA. Four fetal anomalies were selected as case studies: isolated cleft lip, hypoplastic left heart (HLH), spina bifida and Down syndrome[i]. These examples ensured that discussions

included reference to a range of conditions, affecting physical and intellectual capacity and with impact ranging from functionally minor to lethal[ii].

Two groups were recruited to the study: medical and social care professionals. Medical professionals were consultants working in the fields of obstetrics, fetal medicine, neonatology and paediatrics[iii]. A purposive sampling strategy[iv] was adopted in two NHS sites in the North of England. Fourteen interviews were conducted.

Accessing social care professionals was challenging; a snowball sampling approach was found to be a more appropriate method of recruitment. Nine interviews were conducted. The umbrella term 'social care professionals' includes a range of roles; disability care support workers, both mainstream and special needs teachers and workers involved with facilitating access into the community (both enabling independent living and involvement in everyday activities). All participants had experience of working with people with impairments in a supportive social context[i].

Interviews were transcribed verbatim. An inductive thematic analysis[22] was conducted on the data by LC. This approach allows the generation of themes to come from the data. NVivo software was adopted to support analysis. Analysis was conducted alongside data collection which allowed for the exploration of emergent themes. A random sample of interviews was coded separately by RHG, to provide a qualitative equivalent of inter-rater reliability for the coding framework.

## RESULTS

The data revealed remarkably similar themes within both professional groups despite the very different occupational backgrounds. Thus, the data are presented by theme, rather than by professional group. Professionals' accounts suggested that they wanted to present an acceptable moral self-image, and that their discussions on TOPFA reflected this position. Most participants did not support unquestioningly TOP for certain fetal anomalies, but depending on different ethical and moral arguments, they were able to overcome some objections.

### Restoring normality: can it be fixed?

According to the medical model, the body is likened to a machine that can be fixed.[7] 'Fixing' in this context refers

---

[i] Cleft lip arises when the upper lip fails to develop normally. As well as the presurgery disfigurement, infants may experience problems eating, speaking and hearing.[31] Surgery is available, but scarring is often evident, and more extensive clefts need ongoing input from dental and speech therapists. HLH occurs when the left side of the heart fails to develop. Without major heart surgery, HLH is fatal. Babies require multiple operations during childhood but only 65% survive to age 5 years.[32 33] Spina

bifida is a neural tube defect.[34] Medical intervention includes surgery to close the spina bifida and often to manage hydrocephalus, bowel and bladder interventions and, in many instances, devices to assist ambulation (eg, braces) as well as psychosocial intervention. Down syndrome is a chromosomal anomaly-associated varying degrees of cognitive disability. Improvements in management have resulted in an increase in the survival of affected individuals,[35–37] including those with other associated anomalies.[35]

[ii] HLH is lethal without medical intervention.

[iii] Additional information cannot be provided due to the sensitivity of the subject area and for confidentiality purposes.

[iv] Participants selected and invited to participate due to meeting the inclusion criteria of the study.

to 'correcting' or treating the anomaly to move towards what would be considered 'normal' or palliative. This aspect of biomedicine was important due to the assumptions participants made about automatic enrolment into medical intervention after a diagnosis of fetal anomaly. Subsequent interventions included further testing, to establish the extent of the anomaly and medical treatment after birth. The data suggest that enrolment into medical intervention to 'fix' the anomaly is assumed and unquestioned in most instances. The exception to this was Medical Professional 19 who discussed palliative care as a real option that is not well explored by medical professionals, however, the reliance on expert knowledge was still revealed to be part of this option. Despite this, palliative care is still a treatment path, even if this is not to actively treat the impairment. Enrolment into medical intervention, therefore, is revealed by the data as seemingly the option, regardless of whether there is the possibility of a cure.

Isolated cleft lip was a condition that participants, overall, deemed to be minor. This conclusion was based on the success rate of postnatal surgical intervention.

I personally think that, cleft lip is a fairly minor anomaly… that is treatable, and that has a good outcome. (Medical Professional 20)

I thought it was a joke how can anybody terminate a baby for having a cleft lip… especially the way medical science is now and you can get so much plastic surgery. (Social Care Professional 1)

Participants focused on the possibility of correcting the physical anomaly, concluding that cleft lip can be 'fixed' resulting in a normal life (and therefore an unacceptable or questionable justification for TOPFA). However, within both professional groups, some participants drew on other issues which could justify TOP, including maternal choice, and using Clause C of the HFEA if the pregnancy was under the 24-week threshold.

I think it should be a choice, and they should be given as much information as they possibly can… they should have that option. (Social Care Professional 2)

…if you can terminate a healthy baby just because the mother wants to, I don't see why you can't terminate a baby with a minor abnormality if the mother wants to. (Medical Professional 10)

However, the ability to 'fix' an anomaly was coupled with other factors, which feature in the process of negotiating acceptability in this context. Down syndrome cannot be fixed, yet, for participants, this did not automatically equate to acceptable TOPFA. This may be linked to wider societal condemnation, and coupled with the fact that Down syndrome was not conceptualised with suffering.

You don't suffer with Downs syndrome, Downs syndrome is only a problem to the people around you. (Medical Professional 12)

A baby with Downs syndrome who didn't have any associated physical abnormalities, they didn't have cardiac, or heart or gut defect… I would perceive without any doubt that they're not gonna suffer at all. (Medical Professional 13)

For social care professionals, the concept of 'suffering' was also discussed in comparison to a 'normal' person with a difficult life even without impairment. Medical professionals were not overtly opposed to TOP for Down syndrome but were very concerned about ensuring parents knew the full implications of the anomaly. Many were, however, keen to distance themselves personally from the decision.

I would support parents that wanted to terminate a pregnancy for Down syndrome. I sort of have a view that they should be aware of you know, what Down syndrome is and… a lot of parents with Down babies are you know very grateful for having them. (Medical Professional 17)

## Will there be pain?

Conceptualisations of pain were an important consideration for participants from both professional groups. Participants' threshold between acceptable and unacceptable levels of pain differed, depending on various factors, including the anomaly being discussed, perceptions about length of life gained through the pain received, personal views and personal experiences. HLH was useful in teasing out these thresholds because of the need for surgical intervention for survival.[23] Pain will be a feature of a person affected by HLH, and pain was featured as a justification for TOPFA. The pain of ongoing surgery for HLH was conceptualised differently to the 'one off' surgery for isolated cleft lip for example. The notion of preventing a life filled with painful experiences was a key issue for participants:

If I was absolutely convinced there was an abnormality that was just gonna cause pain and distress and then death you know, at an incredibly young age, whatever that abnormality might be, then, they're the kind of cases that you'd be more convinced that you were absolutely doing the right thing. (Medical Professional 8)

It's very difficult to find where the line is and I think probably… my own line… would be somebody who's in pain that can't be alleviated. (Social Care Professional 14)

HLH was deemed an acceptable reason for TOPFA by some social care professionals. This centred on the impact of the child, the medical interventions and the pain they would have to endure. Others, however, compared treatment for HLH to people who have to have 'heart surgery all the time' (Social Care Professional 4) and thus not an acceptable justification for TOPFA.

Overall, medical professionals were able to negotiate acceptable TOP for HLH. Thus, the HLH case study provided the basis for a more nuanced discussion of pain as a process—what level of pain was acceptable to put a child through, to get them to what was seen as a reasonable quality of life. The focus on a live birth with HLH is on surgical intervention, but this was conceptualised as a permanent feature of an affected person's life as HLH cannot be 'fixed', only corrected in palliative terms. More surgical intervention will be required to sustain life for someone with HLH:

> We're talking about long term, you're talking about palliation, so operations… that achieve a circulation but they do not fix the problem, a heart that operates on one pump, and eventually that will fail in some manner. (Medical Professional 9)

> It's very likely that either the baby won't survive or will need lots of surgery which may have a high chance of not being successful. (Medical Professional 11)

The negotiating process exhibited by the participants seemed to regard certainty of significant pain as something that could straightforwardly justify acceptable TOPFA. A normal life experience did not, for them, feature certainty of significant pain. There was also recognition of the necessity of medical intervention which may not be enough to guarantee long-term survival. Therefore, both medical and social care professionals accepted HLH as a serious anomaly, with TOPFA conceptualised as a legitimate option for most participants. There were exceptions, however, as two social care professionals (4 and 14) raised issues around placing a value on life. For them, it was immoral to deny a chance at life. Not actively intervening was also raised by two medical professionals as a legitimate option. For example, one professional stated:

> I'd have a live born baby, take it home, cuddle it, you know, wait for it to die quietly… which is not the same thing as terminating it but also isn't the same thing as embarking on 35 years of, you know, horribly intensive, invasive medical involvement. (Medical Professional 19)

These participants' accounts suggested that they felt it was not necessarily in the best interests of the baby to intervene, but that parents may also have moral objections to TOPFA. This option of palliative care 'can get you off both hooks', (ie, avoiding the decision to proceed with TOPFA while also preventing the baby from living a life of painful experiences). Palliative care was, therefore, seen as a route through the difficulties while maintaining an acceptable moral position and self-view.

### Is it possible to have a 'normal' life trajectory?

The social construction of contemporary Western society places high value on walking, and wheelchairs often symbolise impairment. The wheelchair cannot be hidden in the same way as, for example, bowel or bladder problems. The data show that a number of considerations influence participants' conceptualisations of the acceptability of TOP for spina bifida encompassing both visible and hidden elements; where the lesion is located, the presence of hydrocephalus and mobility issues.

> They realise that the child would need help with the bowel or walking, then, you know, may need a shunt and those things, then that is unacceptable… but when there's a lower defect, we give them the information. (Medical Professional 5)

Many medical and social care professionals had mixed opinions as to the acceptability of TOP for spina bifida. This variation stemmed from the dichotomy of spina bifida being a serious anomaly with serious consequences, and yet, with professionals speculating that if you asked a person affected by the condition if they would rather have not been born, the answer would likely be no.

> It's very hard for me to stand there and look at someone with spina bifida who's, you know, wheelchair bound, and you know is kind of struggling with life, and say that their quality of life is poor. (Medical Professional 8)

The data showed that many professionals in both groups indicated that they saw spina bifida as an acceptable reason for a TOPFA in some instances, or that they would not deny the parents the right to make that decision. Despite some personal misgivings as individuals, participants negotiated their way through the issues to avoid adopting a position that would deny choice to others;

> …spina bifida, they are like serious physical conditions that that child's quality of life will not able to be the same as any other child, they're not gonna be able to fully enjoy aspects of life that other children do. (Social Care Professional 23)

Many of our participants argued that any impairment would make life more difficult (to varying degrees), but that a positive life experience was still achievable. However, they also noted that the huge impact on the lives of family members should not be ignored. A diagnosis of fetal anomaly was seen as changing the life of the parents and siblings forever, thus affecting their 'normal' life trajectory. For example, the level of thought that needs to go into simple aspects of everyday life was discussed by social care professionals, and the inconvenience of unpredictability by medical professionals.

> You can't just hop to the supermarket, and all nip down, you have to plan things around, is it gonna be a long walk, are we gonna be able to park the car closer, just simple things too, have we got pee bags, have we got pads, are they gonna need that. (Social Care Professionals 2)

> …that's what life is like, it's gonna be unpredictable, you're gonna be bringing them in on Christmas day or, you know, you'll plan a holiday and then your

child will be ill, there's all sorts of things that happen. (Medical Professional 20)

The absence of a 'normal' life, however, did not always lead to negotiation of acceptable TOPFA. Down syndrome is not an impairment that can be hidden, unless the affected person is removed from society. In the UK, routine screening for Down syndrome is offered to all pregnant women.[24] The availability of a 'routine test' may in itself be a factor, reinforcing the view that TOPFA is a socially acceptable, widely available option. Like spina bifida, Down syndrome has been discussed as a serious anomaly with serious implications, however negotiating acceptable TOPFA with a moral justification proved difficult for some participants in both professional groups.

…obviously, it (Down syndrome) makes their life more difficult, but there are people that have difficult lives all the time, it doesn't mean they shouldn't have a life. (Social Care Professional 4)

You don't suffer with Down syndrome, Down syndrome is only a problem to the people around you. (Medical Professional 12)

The positive experiences of those affected meant that some participants experienced difficulty in negotiating acceptable TOP for Down syndrome, and many concluded that Down syndrome was at the least a questionable rationale for TOPFA. However, for those who were able to negotiate TOPFA as an acceptable outcome, the issue of societal condemnation was raised as a relevant factor, despite research suggesting that most women diagnosed with Down syndrome opt for TOPFA.[25] These professionals felt that the representations of Down syndrome in mainstream culture tended to reflect the positive experiences of those with Down syndrome and neglected the more negative experiences.

## DISCUSSION

This qualitative study found that both medical and social care professionals adopt classificatory practices which allow them to negotiate a view that TOPFA is an acceptable option, while maintaining a self-image they deem to be morally acceptable. These practices are not dissimilar despite the distinct professional groups, and their different levels of experience with: (1) decision-making that leads to TOPFA or (2) living with impairment. The similarities in their processes of negotiation may be a result of them being shaped by commonly held social understandings of both impairment, TOP and TOPFA. Through discussions of fixing, pain and normal life expectations, professionals were able to negotiate instances of acceptable TOPFA while maintaining a self-image they deem acceptable morally. Thus, they navigate their way between the perceived seriousness of the anomaly in question, perceived immorality of denying choices, and the felt unacceptability of TOPFA as a whole and for particular conditions. Those who indicated that

TOP(FA) should still be an option raised several justifications for their position: (1) using Clause C if the pregnancy is under 24 weeks' gestation, therefore removing the anomaly as the primary justification for TOP and (2) placing heavier emphasis on maternal choice, by framing the denial of choice to women as being immoral. These reasons enabled professionals to either openly reject TOP for particular anomalies, or integrate additional moral arguments into the discussion that allowed them to accept TOP as a legitimate option.

This research has shown how professionals come to decisions about their views on acceptability in relation to TOPFA. The lack of a definition or consensus of terms such as 'substantial risk' and 'serious handicap' that determine whether TOPFA is legally permissible have been raised by others.[2–4] However, it is unlikely that a more focused definition would be welcome,[3] as this would remove the ability to negotiate additional considerations as part of the decision-making process. The lack of categorical definitions does, however, lead to interpretation and means decisions are open to subjective beliefs.[26] Given this decision is arguably based on a great number of complex factors, it is reasoned that it is not possible to have a 'one correct way' to assist parents in making this decision.[27] Our study findings have also shown that despite the presence of a serious anomaly, such circumstances do not automatically equate to a straightforward conceptualisation of TOPFA as acceptable. This is due to perceptions on suffering, pain and broader quality of life. Healthcare is often evaluated considering quality of life,[28] yet there is no definition as to what this means. Those who are suffering from severe disease do not always report having a low quality of life,[28] as indicated by both professional groups, in particular social care professionals. This suggests that individual experience and expectations are considered, alongside other aspects of the anomaly, in how professionals made sense of the concept of quality of life. Thus, the negotiation of acceptable TOPFA, necessarily factors in things outside of the individual anomaly itself and its biological impact. One example was the considerable impact on the whole family; acceptable TOPFA could be negotiated if the TOPFA was in the best interests of the family unit, which may include other children who may also suffer and miss out on a normal life experience because of the fetal anomaly. Thus, an understanding of what it means to live with impairment and disability is key in decision-making processes, something medical professionals involved in TOPFA are arguably less experienced with. For example, assumptions might be made about perceived burden,[29] or that the presence of a disability automatically equates to being 'unhealthy'.[20]

Parents have also been found to have questions and concerns that are not addressed during counselling, in part due to the positioning of counselling within the medical model paradigm.[30] The inclusion of social care professionals in this study will contribute to an increased understanding of how TOPFA is conceptualised because of the different contexts in which social care and medical

professionals work. Comparing the views from individuals across the two professional groups is valuable because disabilities feature as a possible future in the work of one group, and a lived experience in the work of the other. These contrasting standpoints are important to note, given the similarity of the views described by individuals across the two professional groups. This similarity, despite quite different work experiences, may suggest that our participants interpret their differing work experiences with reference to a shared societal-wide acceptance of TOPFA and women's choices.

We propose that, for our participants, the conceptualisation of an acceptable TOPFA decision was influenced by three key factors: whether a particular anomaly can be 'fixed' under the paradigm of biomedicine; what pain this 'fixing' will involve, recognising that medical intervention can be painful; and whether it is reasonable to anticipate that the affected baby could have a 'normal' life trajectory—whether that be in terms of length of life or life years with a meaningful degree of participation and fulfilment. Each of these elements played a part in the ways that participants explained their understandings of the extent to which TOPFA was a morally acceptable option. The knowledge, meanings and interactions the different professional groups gain from their professional roles help shape their perspectives on TOPFA. The level of similarity may also be important in terms of assessing the extent to which knowledge about impairment, normality and suffering is constructed with reference to societal-level factors.

This paper makes no claims to generalisability, especially given the findings have been collected in one geographical location. This research will add to ongoing discussions around TOPFA from the medical professional perspective. This is important given the first point of contact for many parents after a diagnosis of fetal anomaly is in a healthcare setting. While snowball sampling has issues regarding bias,[30] it proved to be invaluable as a recruitment source in this research.

The main contribution from our analysis stems from the comparison between accounts from two different professional groups, where the respective roles of disability in the working environments provide the possibility of a comparative analysis. However, in addition to this main aim, our analysis also contributes to the body of knowledge on each of the two professional groups. The existing knowledge on understandings of TOPFA in the two groups differs, so our work makes a slightly different contribution to each. While medical professionals have been studied previously in relation to TOPFA, it is important to continue research to ensure that: (1) the evidence base remains up to date, especially in a continually changing society and (2) that the evidence available is used to inform effective guidelines that can work with existing clinical practices. Social care professionals are under-represented in research relating to TOPFA, as well as social policy discussions, despite their knowledge and professional experience with people with impairments.

This paper offers perspectives of both these two professional groups, each associated with distinctive and different experiences. Despite these occupational differences, the results showed their views to be remarkably similar when considering acceptable TOPFA. This may suggest a greater influence of societal-wide views, which may need to be considered in any future research.

**Acknowledgements** The authors would like to thank all the study participants for giving up their time to be interviewed.

**Contributors** LC conceived and designed the research study, with RHG, SCR and JR. LC was responsible for the acquisition of data. LC coded and analysed all transcripts with RHG who coded a random sample. All authors were involved in the interpretation of the data. LC wrote the first draft of the manuscript and all authors were involved in subsequent revision. All authors approve the final manuscript.

**Funding** This research was funded by a medical research council/economic and social research council integrated studentship, G0800128-3/1.

**Competing interests** None declared.

**Patient consent** Not required.

**Ethics approval** A favourable ethical opinion was received from the Newcastle and North Tyneside 2 Research Ethics Committee (10/H0907/50).

**Provenance and peer review** Not commissioned; externally peer reviewed.

**Data sharing statement** This research is a result of a PhD by LC. This PhD is available via the Newcastle University depository.

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
