## [Reviewer comments · BMJ Open]

ARTICLE DETAILS

TITLE (PROVISIONAL)	Negotiating acceptable termination of pregnancy for non-lethal fetal anomaly: A qualitative study of Professional perspectives.
AUTHORS	Crowe, Lisa; Graham, Ruth; Robson, Stephen; Rankin, Judith

VERSION 1 – REVIEW

REVIEWER	Dr Robyn Lotto Liverpool John Moores University UK
REVIEW RETURNED	07-Dec-2017

GENERAL COMMENTS	Thank you for inviting to me to review this interesting and relevant work. I believe that it offers a novel insight into decision-making around termination of pregnancy. However, there are aspects of the article that would benefit from some further work. Abstract There are some aspects of the abstract that are confusing, in particular around the two sample groups. This could do with clarifying. Objective – this could do with rephrasing in a more concise manner. For example: This study aims to explore the perspectives of professionals around the issue of termination of pregnancy for non-lethal fetal anomaly. Sample – The sample consists of 14 medical and 9 social care professionals Methods – Semi – structured interview etc Results – the themes are the same, whether social or medical professionals, so cut line 33 pg 2 “medical and social care professionals” and just highlight the themes It would be beneficial to have an extra sentence to summarise the differences between the two groups, as that is one of the strengths of the paper. Keywords I’m not convinced that the North East is a useful keyword. Perhaps consider non-lethal congenital anomaly instead? Introduction There are a number of areas in the introduction that don’t read well and sentence structure is more complex than it needs to be. – Reread and simplify where possible (specific examples given below, but not confined to these examples) A definition of a fetal anomaly plus some information on screening and FASP may make this more relevant and understandable to an
---

international audience

Pg 4 line 26 – add a sentence here explaining the difficulties in determining what is substantial or serious

Line 29 - . Be careful quoting Department of Health termination data as this is known to be under ascertained (see recent paper by Morris et al Accuracy of reporting abortions with Down syndrome in England and Wales: a data linkage study, J of Public Health 38 170-74).

Line 36 – grammatical errors – judge the anomaly (not judging) etc Also use of the term judge doesn't sit comfortably ?apply clinical judgement or something similar

Line 43 – Needs rephrasing. Perhaps ?These issues are compounded by the limited in-utero treatments available, thus reducing parental options to either TOPFA or continue with the affected pregnancy

Line 50 – Your justification for the study doesn't follow on naturally from the previous line. As you have two sample groups, it would benefit from some discussion here about why you have chosen these two groups. Currently, the article ignores the social care side and just talks about the medical aspect. Bringing the two together at this point would benefit the article and bring clarity to your subsequent arguments. I appreciate this aspect is discussed pg 5 line 18- pg6 line 20, but it is disjointed and you need to bring it together (you talk about one, then the other, try to combine them).

Lines 22-29 As for abstract, try to simplify and write more concisely.

Methods

Pg 6 line 41 – state here the focus of this paper – how it differs from the larger study.

Line 41, could be written more concisely “ Semistructured interviews were used to collate data exploring professionals perceptions of the complex issues surrounding TOPFA”

Line 45 – a brief explanation of some of these anomalies would benefit readers. In particular HLHS may require some explanation so readers grasp the complexity and palliative nature of any treatment.

Pg 7 line 7 You use the word lethal, yet the paper is discussing non lethal anomalies.

Line 8/9 – participants were provided – could be shortened to consent was sought or cut out completely. Takes away from the flow.

Pg 7 line 14 – you state two professional groups were recruited but you only list one, then come to the other much further down.

Two groups were recruited; medical consultants and social care professionals. Medical consultants were working in the..... etc.

Pg line 28 appropriate method or feasible method?

Line 23-45 This could be condensed significantly. A lot of repetition

Page 8 line 8 – I would suggest cutting this gender sentence. Either you need to highlight why its important or don't include it. You have also said you cant provide further details (footnote), but then do so.

Page 8 Line 18 –

Nvivo doesn't analyse, it manages data and data analysis.

Results – It would benefit discussing how you are going to present your results. You've made an issue of having 2 groups, but then you present findings together. A couple of sentences at the start of the results would be beneficial explaining that the same themes ran through both groups, and how you are going to present the differences.

Pg 9 line 19 – is palliative considered normal? Not sure I understand this sentence.

	Have you some data to demonstrate the assumption of automatic enrolment into treatment. I think this is a particularly interesting point, particularly as it would then conflict with the offer of palliation. I think you are underselling this aspect, as it offers an important insight that isn't documented elsewhere. You come back to palliation on page 14. Try to bring these aspects together, or at least signpost as it currently feels like it jumps. Page 10 line 32 – this section doesn't feel like it fits in here. To me, it feels more about pain, which you come on to next. Pain and suffering would fit together rather than fixing and suffering. Page 12 line 47 – you talk about acceptability of TOP, that is quite a personal judgement. Keeping it on the professional level, I would expect you to talk about whether it fits with the definition of severe (and hence would fall within the requirements for a TOPFA) Page 13 line 8 – I'm not sure I understand this quote – people who have to have heart surgery all the time – what does that mean in the context of HLHS where some patients may need repeat surgery? Page 15 – line 11 – Title do you mean is it possible to have a normal life trajectory? Page 18 line 18 – would make more sense if you added a the “was at the least” rather than at least. Discussion This brings things together nicely, but you need to set the scene better in the background and results so that it flows throughout the article. Line 17 page 19 – Perceived seriousness – it is a perception rather than an objective measure Use of the word allowed – perhaps enabled is a little softer. Page 21 line 15 – this paragraph is slightly contradictory – you talk about the differences in views then suggest the views of the 2 groups are very similar. A bit of rephrasing would help. Page 21 line 34 – again rephrase, this comes across judgemental. Page 22 line 8, perhaps the use of the term transferable is more in line with the qualitative paradigm than generalizable. Page 22 line 13 – Scprof views are under-represented and their inclusion have enabled their views to be represented – rephrase – too many ‘represented’ Page 22 line 27 – I'm not sure about your rationale/justification for this paper – your selling point is looking at the differences and similarities between these two professional groups. That is what is unique, rather than seeking to see if clinician views have changed in the last year or so. You repeat the under-representation of social care professionals several times. This isn't required. Your final paragraph (and in particular the final sentence) sounds like you ran out of steam. It doesn't reflect the impact your paper could have. I would consider rewriting. It would be beneficial to spell out some of the implications. General comments – Look at sentence structure where you've used TOPFA. Eg line 44 page 8 you've written TOPFA for certain fetal anomalies. Rather use TOP for certain fetal anomalies. Other examples throughout text. Overall, this paper has a lot to offer. Some significant reworking throughout is required. However, once complete, I think that the paper will offer a novel insight and open up discussion. Thank you for inviting me to review.
--	--

REVIEWER	Asim Kurjak Medical School University of Zagreb
-----------------	--

	Croatia
REVIEW RETURNED	14-Dec-2017
GENERAL COMMENTS	I read with great interest paper Negotiating acceptable termination of pregnancy for non-lethal fetal anomaly: Professional perspectives. A qualitative study, which was sent to me for review. It is very organized paper about hot topic of the large scientific and public interest. I did not find any reason for changes or adaptation and do like to propose publication of the paper in its present form.

VERSION 1 – AUTHOR RESPONSE

Dear Editor,

Thank you for providing us with reviewers' comments and the opportunity to address them. We have addressed all the comments and provided them in a separate file upload.

Reviewer 1's comments are given in italics with our response underneath.

Reviewer 2: No changes recommended. We thank the reviewer for their positive comment.

Kind Regards

Lisa Crowe, Ruth H Graham, Stephen C Robson, Judith Rankin.

Dear Editor,

Thank you for providing us with reviewers' comments and the opportunity to address them. We have addressed all the comments below; the reviewer's comments are given in italic.

Reviewer 1

1. Abstract:

There are some aspects of the abstract that are confusing, in particular around the two sample groups. This could do with clarifying.

Thank you for the comments relating to the abstract. We have amended the objective in light of your comments but think the additional detail in the rest of the abstract is needed.

It would be beneficial to have an extra sentence to summarise the differences between the two groups, as that is one of the strengths of the paper.

An additional sentence has been added.

2. Keywords

I'm not convinced that the North East is a useful keyword. Perhaps consider non-lethal congenital anomaly instead?

We have removed North East from the keywords. We are reluctant to use congenital anomaly, as this means born with an anomaly, whereas fetal anomaly is specific to a diagnosis during pregnancy, which is the focus of this paper.

3. Introduction

There are a number of areas in the introduction that don't read well and sentence structure is more complex than it needs to be. – Reread and simplify where possible (specific examples given below, but not confined to these examples).

Thank you for your comments. An additional proof read has been conducted by LC and JR.

A definition of a fetal anomaly plus some information on screening and FASP may make this more relevant and understandable to an international audience

The following sentences have been added to the manuscript:

Fetal anomaly screening is offered to all pregnant women through the NHS Fetal Anomaly Screening Programme (FASP). The FASP defines a fetal anomaly as an abnormality which “may indicate the baby might die shortly after birth, conditions that may benefit from treatment before birth, to plan delivery in an appropriate hospital/Centre and/or to optimize treatment after the baby is born” (FASP, 2016: 5).

Pg 4 line 26 – add a sentence here explaining the difficulties in determining what is substantial or serious

Thank you for your comment. We discuss these difficulties on page 20 (discussion). The following sentence has also been added to the introduction also.

“Difficulties have been noted in defining terms such as “substantial” and “serious”.”

Line 29 - . Be careful quoting Department of Health termination data as this is known to be under ascertained (see recent paper by Morris et al Accuracy of reporting abortions with Down syndrome in England and Wales: a data linkage study, J of Public Health 38 170-74).

Thank you for your comment. We have now added the word ‘reportedly’ to acknowledge this known underascertainment.

Line 36 – grammatical errors – judge the anomaly (not judging) etc Also use of the term judge doesn’t sit comfortably ? apply clinical judgement or something similar

Thank you for your comment. We do not feel using clinical judgment is appropriate in this context, as that assumes that clinical judgement is used in all cases, which the data suggests is not the case. We use the concept ‘judge’ because it encompasses clinical judgment alongside other evaluations that are not necessarily based on clinical experience or evidence. Therefore, we feel ‘judge’ is most appropriate in this context.

Line 43 – Needs rephrasing. Perhaps ? These issues are compounded by the limited in-utero treatments available, thus reducing parental options to either TOPFA or continue with the affected pregnancy

Thank you for this suggestion. This has now been incorporated.

Line 50 – Your justification for the study doesn’t follow on naturally from the previous line. As you have two sample groups, it would benefit from some discussion here about why you have chosen these two groups. Currently, the article ignores the social care side and just talks about the medical aspect. Bringing the two together at this point would benefit the article and bring clarity to your subsequent arguments. I appreciate this aspect is discussed pg 5 line 18- pg6 line 20, but it is disjointed and you need to bring it together (you talk about one, then the other, try to combine them).

Thank you for this comment. We do not agree that the article ignores the social care side; given we have been unable to find any research with their views on TOPFA, inevitably, there will be some more discussion on medical professionals given there is existing research on medical professionals in relation to TOPFA. We have done some restructuring to the paragraphs which removes the disjointedness.

Lines 22-29 As for abstract, try to simplify and write more concisely.

We have made the suggested re-wording.

4. Methods

Pg 6 line 41 – state here the focus of this paper – how it differs from the larger study.

Thank you for this comment. This has now been clarified by adding that this paper reports the qualitative data.

Line 41, could be written more concisely “Semistructured interviews were used to collate data exploring professionals perceptions of the complex issues surrounding TOPFA”

This has now been changed.

Line 45 – a brief explanation of some of these anomalies would benefit readers. In particular HLHS may require some explanation so readers grasp the complexity and palliative nature of any treatment.

Thank you for this comment. We have provided a summary in the footnote of each condition.

Pg 7 line 7 You use the word lethal, yet the paper is discussing non lethal anomalies.

Thank you for your comment. This is because hypoplastic left heart is lethal without intervention. A footnote has now been added to explain this.

Line 8/9 – participants were provided – could be shortened to consent was sought or cut out completely. Takes away from the flow.

This statement now been removed.

Pg 7 line 14 – you state two professional groups were recruited but you only list one, then come to the other much further down. Two groups were recruited; medical consultants and social care professionals. Medical consultants were working in the..... etc.

Thank you for your comment. They are discussed in separate paragraphs to allow for explanation of the different recruitment methods adopted. We have re-worded this section slightly in the light of your comments.

Pg line 28 appropriate method or feasible method? Line 23-45 This could be condensed significantly.

A lot of repetition

Thank you for your comment. Re appropriate/feasible: We feel appropriate encompasses the meaning more as feasible suggests the only possible method when that was not the case.

Re line 23-45. Thank you for your comment. We have condensed this paragraph down slightly, but we feel much of the detail is needed to adequately explain the issues being discussed.

Page 8 line 8 – I would suggest cutting this gender sentence. Either you need to highlight why its important or don't include it. You have also said you cant provide further details (footnote), but then do so.

This sentence has now been removed.

Page 8 Line 18 – Nvivo doesn't analyse, it manages data and data analysis.

This sentence has now been amended.

Results – It would benefit discussing how you are going to present your results. You've made an issue of having 2 groups, but then you present findings together. A couple of sentences at the start of the results would be beneficial explaining that the same themes ran through both groups, and how you are going to present the differences.

The following sentence has now been included: "The data revealed remarkably similar themes within both professional groups despite the very different occupational backgrounds. Thus, the data are presented by theme, rather than by professional group."

Pg 9 line 19 – is palliative considered normal? Not sure I understand this sentence.

Palliative is referring to the fact that some anomalies cannot be fixed (HLH) – the only way to no longer have HLH is to have a heart transplant, which is also a palliative care option.

Have you some data to demonstrate the assumption of automatic enrolment into treatment. I think this is a particularly interesting point, particularly as it would then conflict with the offer of palliation. I think you are underselling this aspect, as it offers an important insight that isn't documented elsewhere. You come back to palliation on page 14. Try to bring these aspects together, or at least signpost as it currently feels like it jumps.

Thank you for this comment. Some signposting has now been added.

Page 10 line 32 – this section doesn't feel like it fits in here. To me, it feels more about pain, which you come on to next. Pain and suffering would fit together rather than fixing and suffering.

This section refers to the fact that an anomaly cannot be fixed which doesn't automatically equate to suffering. We think that moving this section would leave a gap in the argument. When we go on to discuss pain, we are referring to pain over a prolonged period, which is discussed as being conceptualised differently to pain from a one off surgery, such as for cleft lip.

Page 12 line 47 – you talk about acceptability of TOP, that is quite a personal judgement. Keeping it on the professional level, I would expect you to talk about whether it fits with the definition of severe (and hence would fall within the requirements for a TOPFA)

Thank you for your comment. We think it is important to keep this in the article as it is important to report that medical professionals can and do use Clause C if the pregnancy is sub 24 weeks. This is used when the parents may request TOPFA but the professional does not agree with that fetal anomaly in question is serious enough to justify TOPFA. We feel this is useful to keep in the text as it

highlights that whilst the legal category for TOP may not be applicable, the issue of anomaly still features in individual discussions of rationales for TOP in more complex ways.

Page 13 line 8 – I'm not sure I understand this quote – people who have to have heart surgery all the time – what does that mean in the context of HLHS where some patients may need repeat surgery?

A slight amendment to the sentence has been made.

Page 15 – line 11 – Title do you mean is it possible to have a normal life trajectory?

The 'living a good life' part of this title has now been removed to avoid any confusion.

Page 18 line 18 – would make more sense if you added a the "was at the least" rather than at least. This has now been amended.

5. Discussion

This brings things together nicely, but you need to set the scene better in the background and results so that it flows throughout the article.

The changes made through the other comments have addressed this issue.

Line 17 page 19 – Perceived seriousness – it is a perception rather than an objective measure Use of the word allowed – perhaps enabled is a little softer.

This change has been made.

Page 21 line 15 – this paragraph is slightly contradictory – you talk about the differences in views then suggest the views of the 2 groups are very similar. A bit of rephrasing would help.

Some rephrasing has been done on this paragraph.

Page 21 line 34 – again rephrase, this comes across judgemental.

Some rephrasing has been done on this paragraph.

Page 22 line 8, perhaps the use of the term transferable is more in line with the qualitative paradigm than generalizable.

Thank you for your comment. Our findings are not generalisable but that doesn't mean they cannot be transferred and adopted in discussions. We would prefer to keep with this term.

Page 22 line 13 – Scprof views are under-represented and their inclusion have enabled their views to be represented – rephrase – too many 'represented'

This sentence has now been removed as per a later suggestion.

Page 22 line 27 – I'm not sure about your rationale/justification for this paper – your selling point is looking at the differences and similarities between these two professional groups. That is what is unique, rather than seeking to see if clinician views have changed in the last year or so.

Thank you for this comment. Some additional writing has been added to further clarify this.

You repeat the under-representation of social care professionals several times. This isn't required.

This has now been amended.

Your final paragraph (and in particular the final sentence) sounds like you ran out of steam. It doesn't reflect the impact your paper could have. I would consider rewriting. It would be beneficial to spell out some of the implications.

Thank you for this comment. Some re-writing has been done on the final paragraph.

General comments – Look at sentence structure where you've used TOPFA. Eg line 44 page 8 you've written TOPFA for certain fetal anomalies. Rather use TOP for certain fetal anomalies. Other examples throughout text.

This has now been amended.

Overall, this paper has a lot to offer. Some significant reworking throughout is required. However, once complete, I think that the paper will offer a novel insight and open up discussion.

Thank you for your positive evaluation of our paper.

Reviewer 2: No changes recommended.

We thank the reviewer for their positive comment.

Kind Regards

Lisa Crowe, Ruth H Graham, Stephen C Robson, Judith Rankin.

VERSION 2 – REVIEW

REVIEWER	Dr Robyn Lotto Liverpool John Moores University Liverpool UK
REVIEW RETURNED	23-Jan-2018
GENERAL COMMENTS	Thank you for inviting me to review this article. I greatly enjoyed reading it, and appreciate the work gone into revising the article. Whilst I would suggest that this paper is now accepted, there appears to be a formatting issues with footnotes 1 and 2, where the numbers don't appear to reflect the footnotes themselves. Please reposition. No further comments.